# Recent Advances of m6A Demethylases Inhibitors and Their Biological Functions in Human Diseases

**DOI:** 10.3390/ijms23105815

**Published:** 2022-05-22

**Authors:** Yazhen You, Yundong Fu, Mingjie Huang, Dandan Shen, Bing Zhao, Hongmin Liu, Yichao Zheng, Lihua Huang

**Affiliations:** 1Green Catalysis Center, College of Chemistry, Zhengzhou University, Zhengzhou 450001, China; youyz2020@163.com (Y.Y.); fudcqupt@163.com (Y.F.); h18836930469@163.com (M.H.); 2School of Pharmaceutical Sciences, Zhengzhou University, Zhengzhou 450001, China; summersdd@163.com (D.S.); zhaobing@zzu.edu.cn (B.Z.); liuhm@zzu.edu.cn (H.L.); yichaozheng@zzu.edu.cn (Y.Z.)

**Keywords:** m6A demethylases, FTO, AKBH5, human diseases, inhibitors

## Abstract

N6-methyladenosine (m6A) is a post-transcriptional RNA modification and one of the most abundant types of RNA chemical modifications. m6A functions as a molecular switch and is involved in a range of biomedical aspects, including cardiovascular diseases, the central nervous system, and cancers. Conceptually, m6A methylation can be dynamically and reversibly modulated by RNA methylation regulatory proteins, resulting in diverse fates of mRNAs. This review focuses on m6A demethylases fat-mass- and obesity-associated protein (FTO) and alkB homolog 5 (ALKBH5), which especially erase m6A modification from target mRNAs. Recent advances have highlighted that FTO and ALKBH5 play an oncogenic role in various cancers, such as acute myeloid leukemias (AML), glioblastoma, and breast cancer. Moreover, studies in vitro and in *mouse* models confirmed that FTO-specific inhibitors exhibited anti-tumor effects in several cancers. Accumulating evidence has suggested the possibility of FTO and ALKBH5 as therapeutic targets for specific diseases. In this review, we aim to illustrate the structural properties of these two m6A demethylases and the development of their specific inhibitors. Additionally, this review will summarize the biological functions of these two m6A demethylases in various types of cancers and other human diseases.

## 1. Introduction

Existing epigenetics mainly involve reversible chemical modifications of DNA, histone, and RNA, which can be inherited through cell division without changing the DNA sequence. So far, over 100 kinds of post-transcriptional modifications of RNA have been defined, including N1-methyladenosine (m1A), N6-methyladenosine (m6A), and 2-O-dimethyladenosine (m6Am) [1,2]. An important discovery was that m6A was the most abundant in RNA internal modifications between different species [1,3], and was also presented in eukaryotic mRNAs [4] and noncoding RNAs [5]. As with any other epigenetic modifications, m6A methylation can be individually dynamically installed; removed; and recognized by the so-called “writers”, “erasers”, and “readers”, individually (Figure 1) [6,7]. The occurrence of m6A methylation is controlled by a core methyltransferase complex, i.e., the “writers”, composed of several core proteins, including methyltransferase-like 3 and 14 (METTL3 and METTL14), and wilms tumor 1-associated protein (WTAP) [8]. Only two m6A demethylases “erasers”, fat-mass- and obesity-associated protein (FTO) [9] and alkB homolog 5 (ALKBH5) [10], have been found so far, which can specifically eliminate the m6A sites from target mRNAs. The “readers”, including YT521-B homology (YTH) domain family 1–3 (YTHDF1-3), YTH domain containing 1–2 (YTHDC1-2), insulin-like growth factor 2 mRNA-binding proteins (IGF2BPs) (including IGF2BP1–3), and eukaryotic initiation factor 3 (EIF3), control the fate of the target mRNA by recognizing and binding to m6A sites [11]. Notably, these ‘readers’ perform different functions after binding to RNA containing m6A, resulting in different destinies for target RNA [12]. For example, YTHDF1 enhances RNA translation while YTHDF2 induces degradation of the transcripts.

In recent decades, epigenetic studies have explicitly highlighted the relationship between m6A demethylases and RNA metabolism, which can affect gene expression and animal development, as well as human disease progression [6,8,9]. FTO/ALKBH5 removes m6A modification with the assistance of cofactors 2-oxoglutarate (2OG) and Fe^2+^. They all belong to the AlkB subfamily of the 2OG dioxygenase superfamily [13,14]. Structurally, both FTO and ALKBH5 contain a highly conserved double-stranded β-helix (DSBH) fold (also called jelly-roll motif), providing a scaffold for the conserved ferrous ion (HXD/E…H motif) and 2OG binding site (Figure 2A–C). Moreover, structural insights into FTO and ALKBH5 identified some active-site residues that are critical to substrate-binding specificity and selectivity. These specific residues may help to design selective inhibitors against other AlkB family members.

We performed extensive research on the databases of PubMed, Google Scholar, Web of Science, and SciFinder to extract literatures from 1974 to 2021. The keywords used included: “m6A demethylases”; “FTO”; “AKBH5”; “inhibitors”; “crystal structure”; “human diseases”; “cancer”; and “therapy responses”. Moreover, we extracted protein crystal structures from the protein data bank (RCSB PDB) database, and then the Molecular Operating Environment (MOE, 2019.0102) software was used to generate two-dimensional (2D) and three-dimensional (3D) ligand–protein interactions diagrams. This review will discuss and compare the structural characteristics of FTO and ALKBH5 according to recent literature. Additionally, this review will summarize the advances of their inhibitors and their functions in various biological processes and diseases.

## 2. Structures of Fat-Mass- and Obesity-Associated Protein (FTO) and alkB Homolog 5 (ALKBH5)

### 2.1. Structures and Functions of FTO

The complete human FTO protein contains 498 amino acids, mainly composed of an N-terminal domain (NTD, residues: 1–326) and a C-terminal domain (CTD, residues: 327–498) (Figure 2A,B). Han et al. firstly reported a crystalline complex of FTOΔ31 and 3-methylthymidine (3-meT) (PDB ID: 3LFM), where 3-meT was a mononucleotide substrate for FTO (Figure 2B) [15]. They found an interaction between the CTD and NTD of FTO, which contributed to stabilizing NTD and promoting the catalytic activity of NTD. Like other AlkB family members, the active center of FTO contained a conserved DSBH (β5-β12) domain located in its NTD. Structural analysis showed that two α-helixes supported DSBH on one side, and the other side was covered by the L1 loop (a long loop including residues 213–224) (Figure 2B). Notably, structural data indicated that this L1 loop was highly conserved across different FTO species and was involved in substrate selection. Specifically, the structural comparison showed that the L1 loop and the unmethylated strand of the DNA double strand competed to bind to FTO, implying that FTO preferred single-stranded nucleic acids as catalytic substrates.

### 2.2. Structures and Functions of ALKBH5

To date, several crystalline structural features of *human* and *zebrafish* ALKBH5 with different ligands have been resolved [16,17,18,19]. The full-length human ALKBH5 protein contains 394 amino acids and consists of two components: NTD and CTD (Figure 2A). Unlike FTO, the absence of the CTD (residues 293–394) of ALKBH5 has no significant effect on m6A demethylase activity. In the truncated construct of ALKBH566–292 (PDB ID: 4NJ4) by Aik and colleagues, the conserved DSBH core fold was constituted by eight anti-parallel β-strands, forming two β-sheets: the major β-sheet and the minor β-sheet (Figure 2C) [16]. Importantly, several research groups have identified two major structural features of ALKBH5 [16,17,18]. A significant structural feature of ALKBH5 is located on the “nucleotide recognition lid (NRL)”. The NRL region of ALKBH5 included two amino acid peptide chains (named motif 1 and motif 2), which dynamically assisted in recognizing nucleic acid substrates. Interestingly, while motif 1 of ALKBH5 left a larger open space above the active center, the long motif 2 was more flexible and had undergone a conformational change after interacting with the nucleic acid substrate. Once bonded to the substrate, motif 2 flipped up to the open area exposed by motif 1 to accommodate it. Another significant structural feature was the unique disulfide bond between residues Cys230 and Cys267. In particular, this disulfide bond restricted the conformation of the L2 loop related to the L1 loop of FTO, thereby preventing the double-stranded substrate from entering the ALKBH5 active center. Together, these structural insights demonstrated the importance of structural features in maintaining the substrate specificity of ALKBH5.

### 2.3. Structural Comparisons of FTO and ALKBH5

Even though they shared the same catalytic mechanism and DSBH active domain, FTO and ALKBH5 exhibit several differences in substrate specificity and small-molecule inhibitor selection. The identification of some conserved binding residues partially explained the functional differences between FTO and ALKBH5. The most notable one was that they both had conserved nucleic acid binding residues that individually exhibited different affinity and selectivity to the substrate. Therefore, we used MOE software to generate two 3D protein–ligand interaction diagrams for crystal complexes of FTO and ALKBH5 (PDB ID: 3LFM and 4NRO), which displayed the binding information between the ligand and the protein. Creating 3D protein-ligand interaction diagrams consisted of the following steps: (1) open the Visualization Setup panel and change the theme to “Standard White” directly; (2) click “Site View” to isolate the ligand and pocket in 3D; (3) press “Ribbon”, “Style”, and “×” to hide the protein backbone ribbon; (4) click “Render”, “Atoms”, and “Residue” to display residue names; (5) click “Render”, “Atoms” to change the color and display mode of the ligand, residues, and metal ion; (6) click “Effects & Text” in the Visualization Setup panel and adjust the “Text Size” to 1; and (7) click “Save Picture” from the bottom of the Visualization Setup panel in the TIF format. In the crystalline complex of FTOΔ31-3-meT (PDB ID: 3LFM), residues Tyr108 and His231 sandwiched the nucleobase ring of 3-meT, while Leu109 and Val228 packed the sugar ring by hydrophobic interactions. Likewise, the residue Tyr141 in ALKBH5 was determined, which corresponded to the Tyr108 in FTO. Further, there were three hydrogen bonds between FTO and 3-meT in the crystalline complex of FTOΔ31-3-meT. Two of them (O2-3-meT and the conserved Arg96; O4-3-meT and the amide nitrogen of Glu234) were vital for FTO selection against differently methylated nucleobases. In line with Arg96 of FTO, residues Arg130 and Lys132 of ALKBH5 that might interplay with m6A contributed to its higher binding affinity and specificity for m6A. Additionally, the residue Phe234 from the L2 loop of ALKBH5 was indispensable in flipping the m6A base into the active site. Furthermore, the active pockets of FTO and ALKBH5 carried conserved residues in combination with metal ions and 2OG cofactors. In FTO, three conserved residues (His231, Asp233, and His307) of the key HX(D/E) motif directly coordinated to Fe^2+^ (Figure 2D). Similarly, in the crystalline structure of ALKBH5 (PDB ID: 4NRO), Mn^2+^ was octahedrally coordinated by three residues (His204, Asp206, and His266) from the DSBH side chains of ALKBH5, water molecules, and a cofactor (Figure 2E). Apart from chelating iron ions, the 2OG cofactor also interacted with multiple residues in the functional pockets for different AlkB members. For example, they were Asn193, Tyr195, Lys132, Arg283, and Arg277 in ALKBH5, while they were Arg316, Ser318, Tyr295, Asn205, and Arg96 in FTO. Notably, crystallographic and biochemical studies revealed a much smaller active cavity for ALKBH5 than that for FTO (490.2 Å3 for ALKBH5 and 817.5 Å3 for FTO) [17]. Therefore, ALKBH5 inhibitors may require smaller-sized compounds. In short, the structural analysis of FTO and ALKBH5 and the evaluation of their substrate-binding specificity and selectivity paved the way for the study of catalytic activity and the design of selective inhibitors.

## 3. FTO and ALKBH5 in Human Diseases

### 3.1. FTO Is Involved in Regulating Various Diseases

#### 3.1.1. FTO in Adipogenesis and Metabolism-Related Diseases

Initially, FTO was reported to link to obesity in the genome-wide search for type 2 diabetes-sensitive genes [20]. Since obesity was an accepted risk factor for some common diseases, it was reasonable to assume that FTO was a factor contributing to Human Diseases. Subsequently, cumulative studies have confirmed this hypothesis. FTO was markedly associated with many disease risks, such as leukemia [21], cardiovascular disease [22], breast cancer [23], melanoma [24,25], and endometrial cancer [26].

As described in more recent research, FTO inactivation may prevent obesity, while over-expression of FTO induced the expression of ghrelin that controls intake behavior [27,28]. Further, FTO participated in adipogenesis by mediating mRNA production of adipogenesis regulatory factors. For instance, FTO promoted adipocyte differentiation by controlling RNA splicing of Runt-related transcription factor 1 (RUNX1T1, an adipogenesis-related transcription factor) in an m6A-erase manner [29]. Peng et al. recently validated that treating hepatic cells with FTO inhibitor entacaponesignificantly decreased the expression of forkhead box protein O1 (FOXO1) by inhibiting the m6A demethylase activity of FTO [30]. Inspired by previous studies that found that FOXO1 was crucial for hepatic gluconeogenesis in the fasting condition, they evaluated the specific effect of the FTO-FOXO1 axis in *mice* [31]. Intriguingly, an in vivo experiment demonstrated that FTO deficiency decreased body weight and fasting blood glucose concentration in diet-induced obese *mice* by acting on the FTO-FOXO1 regulatory axis.

Krüger et al. recently verified the relationship between FTO and obesity-induced metabolism as well as vascular changes [32]. Endothelial FTO knockout protected *mice* from obesity-induced insulin resistance, hyperglycemia, and hypertension in endothelial cells and skeletal muscle in obese conditions. The loss of FTO increased m6A level and stabilized lipocalin-type prostaglandin D synthase (L-PGDS) mRNA, thus upregulating its expression. Subsequently, L-PGDS promoted the synthesis of prostaglandin D2 (PGD2), and PGD2 stimulated AKT phosphorylation in endothelial cells. In parallel to previous reports, obesity decreased AKT phosphorylation in endothelial cells, impairing glucose and insulin tolerance [33]. These results indicate that FTO played a metabolic and vascular role independent of the gene’s role in obesity. Therefore, the selective inhibition of FTO may be used to treat dysregulated metabolic homeostasis.

#### 3.1.2. FTO in Heart Failure

The association between FTO-mediated m6A demethylation and cardiovascular disease, including heart failure, has also drawn the attention of researchers. Berulava et al. reported that genetic ablation of FTO can lead to an accelerated progression of heart failure [34]. Consistently, Mathiyalagan and co-workers found decreased expression of FTO in failing hearts, which resulted in decreasing cardiomyocyte contractile function [35]. Both sustained and transient overexpression of FTO attenuated ischemia-induced cardiac dysfunction in *mouse* models of myocardial infarction. Further studies revealed that FTO overexpression prevented the degradation of cardiac contractile transcripts and improved their protein expression under ischemia conditions through selective demethylation. More recently, a study revealed that FTO could alleviate cardiac dysfunction in *mice* with pressure overload-induced heart failure by regulating glucose uptake and glycolysis through facilitating the expression of glycolysis-related genes such as phosphoglycerate mutase 2 (PGAM2) [36]. These findings suggest that FTO may be a therapeutic target for heart failure treatment.

#### 3.1.3. FTO in Leukemia

More recent studies confirmed that FTO was associated with an oncogenic effect in acute myeloid leukemias (AML) [37,38]. Importantly, Li et al. discovered that FTO was highly expressed in subtypes of AML cells [37]. AML is a disease driven by a small proportion of leukemic stem cells (LSCs), and the persistence of LSCs is considered to be a principal cause of disease recurrence. Functionally, FTO predominantly played its oncogenic role in promoting leukemic oncogene-mediated cell transformation and leukemogenesis. In particular, FTO suppressed the expression of tumor suppressors, including ankyrin repeat and SOCS box containing 2 (ASB2) and retinoic acid receptor alpha (RARA) by eliminating m6A sites. Further research confirmed that FTO suppressed all-trans retinoic acid (ATRA)-induced leukemia cell differentiation by reducing ASB2 and RARA expression [37]. Subsequently, Su et al. found that R-2-hydroxyglutarate (R-2HG) (Figure 3) displayed an anti-leukemic activity by inhibiting FTO demethylase activity, thus altering the expressions of MYC and CCAAT/enhancer-binding protein alpha (CEBPA) [39]. In R-2HG-sensitive leukemic cells, treating leukemic cells with R-2HG significantly upregulated the m6A levels of MYC/CEBPA mRNA by blocking the binding of FTO to mRNA targets. Later, it was found that YTHDF2 specifically recognized the increased m6A modification, thereby weakening MYC/CEBPA transcripts to trigger their degradation. This study demonstrated that R-2HG could effectively inhibit FTO demethylase activity, thereby reducing the proliferation/survival of leukemic cells with highly expressed FTO by modulating m6A/MYC/CEBPA signaling [39].

Shortly after tyrosine kinase inhibitors (TKIs) were used in the clinical treatment of leukemia, it was found that rapidly acquired resistance to TKIs became a significant obstacle. Yan et al. revealed that the formation of a drug-resistant phenotype during TKI therapy corresponded to the over-expression of FTO in leukemic cells [40]. When FTO was inactivated by gene knockdown or chemical inhibitors, resistant cells became sensitive again to TKIs, accompanied by the MER proto-oncogene, tyrosine kinase (MERTK), and B-cell lymphoma-2 (Bcl-2) reduction. Furthermore, they confirmed that when FTO demethylase activity was inhibited with rhein (Figure 3), combined with TKIs, it greatly inhibited tumor growth in nude *mice*. These findings suggested that the inhibition of FTO by small-molecule compounds might be an effective approach to treating leukemia.

#### 3.1.4. FTO in Glioblastoma

Cui et al. recently confirmed that reversible m6A modification in mRNA was crucial for the tumor-promoting role of glioblastoma stem cells (GSCs), mainly by maintaining self-renewal capacity and improving tumorigenesis [41]. Both knockdown METTL3/METTL14 and over-expressed FTO could facilitate GSC growth and self-renewal by reducing the overall level of m6A modification. In particular, they found that decreased m6A modification directly led to increased expressions of oncogenes, such as A disintegrin and metalloproteinase 19 (ADAM19), ephrin type-A receptor 3 (EPHA3), and Krüppel-like factor 4 (KLF4). In addition, in vivo experiments showed that inhibiting FTO activity with FTO inhibitors MA2 (Figure 4) significantly repressed GSC-initiated tumorigenesis and prolonged the lifespan of GSC-grafted *mice*. 

#### 3.1.5. FTO in Breast Cancer

FTO plays a pro-tumor role in *human* breast cancer [42]. In the previous studies, several FTO single-nucleotide polymorphisms (SNPs) showed an association with breast cancer risk [23,43,44], and FTO is highly expressed in *human* breast cancer tissues. The inhibition of FTO by MO-I-500 (Figure 3) reduced the proliferation of breast cancer cells, whereas the over-expression of FTO significantly promoted breast cancer progression [45,46]. Additionally, Niu and colleagues reported that FTO exerted its pro-tumor activity by downregulating the expression of Bcl-2 nineteen kilodalton interacting protein 3 (BNIP3), a pro-apoptotic member of the Bcl-2 family of apoptotic proteins [46]. As described in a recent study, BNIP3 functioned as a tumor suppressor in breast cancer by inducing cell apoptosis [47]. In this case, FTO erased m6A methylation in the 3′-UTR of BNIP3 mRNA, resulting in the degradation of BNIP3. Hence, we propose that FTO may be an effective target for the treatment of breast cancer.

#### 3.1.6. FTO in Melanoma

Earlier findings from 2013 suggested that FTO genetic variations were associated with an increased risk of melanoma [24,25]. More recently, Yang et al. examined the regulation of FTO as m6A demethylase on malignant melanoma samples and multiple melanoma cell lines [48]. Mechanistic studies showed that FTO played a pro-tumor role in vitro and in vivo. Primarily, they found that FTO could be upregulated in response to metabolic stress and starvation via autophagy and nuclear factor kappa B (NF-κB) pathways. In addition, FTO suppression increased the m6A methylation of critical pro-tumorigenic melanoma cell-intrinsic genes (programmed cell death-1, PD-1; C-X-C motif chemokine receptor 4, CXCR4; and sex-determining region Y-box 10, SOX10). After that, the m6A reader YTHDF2, which mainly mediated RNA degradation, significantly reduced their expression. In addition, FTO displayed an effect on anti-PD-1 resistance. The inactivation of FTO reduced the drug resistance to anti-PD-1 treatment of melanoma in *mice*. These results suggest that the inhibition of FTO activity combined with anti-PD-1 could be an effective strategy in immunotherapy.

#### 3.1.7. FTO in Endometrial Cancer

In previous years, the inherent link between FTO and endometrial cancer had also attracted the attention of scientists. Researchers examined the expression of FTO in endometrial tumor tissues, and immunohistochemistry staining showed that FTO was highly expressed in endometrial cancer tissues [49,50]. Mechanism studies show that β-estradiol (E2) increased FTO expression and subsequently promoted proliferation and invasion phenotypes by activating the PI3K/AKT and MPAK signal pathways [49]. Moreover, estrogen was beneficial for FTO nuclear localization through the mTOR signaling pathway dependent on estrogen receptor α (ERα) [50]. More recent work by Zhang’s group found that FTO promoted endometrial cancer metastasis by activating the Wnt signaling pathway [51]. Mechanistically, FTO blocked YTHDF2-mediated mRNA degradation by eliminating m6A methylation in the 3′-UTR regions of homeobox B13 (HOXB13) mRNA, thereby increasing HOXB13 expression. As a result, high expression of HOXB13 activated the Wnt signaling pathway leading to tumor metastasis and invasion. These findings suggested the possible use of FTO as a therapeutic target for endometrial cancer.

#### 3.1.8. FTO in Gastric Cancer

A study published in 2017 has demonstrated that FTO was highly expressed in gastric cancer cell lines and correlated to poor prognosis in patients with gastric cancer [52]. More recently, by performing a univariate Cox regression analysis on the expression levels in the Cancer Genome Atlas (TCGA) dataset, Su et al. found that high FTO expression was related to poor survival of gastric cancer patients [53]. Likewise, another report claimed that reduced m6A modification contributed to malignant phenotypes in gastric cancer [54]. Mechanistic studies showed that knockdown of METTL14 suppressed m6A modification, thereby promoting the proliferation and invasion of gastric cancer cells and activating the Wnt/PI3K-Akt signaling. In contrast, these phenotypic and molecular changes could be attenuated by upregulation of m6A content through knockdown of FTO. Moreover, very recent work by Yang et al. demonstrated that histone deacetylase 3 (HDAC3) improved gastric cancer progression by affecting the forkhead box transcription factor A2 (FOXA2)-mediated FTO-m6A-MYC axis [55]. In vitro experiments showed that HDAC3 facilitated the proliferation, migration, and invasion of gastric cancer cells by inhibiting the expression of FOXA2. Mechanism studies confirmed that FOXA2 accurately bonded to the promoter region of FTO, which significantly inhibited FTO transcription and expression. On the contrary, FTO stabilized the mRNA stability of MYC by eliminating m6A methylation, thus increasing its expression. The further investigation highlighted that depletion of HDAC3 impeded tumor growth and reduced the protein synthesis of FTO/MYC, and elevated the expression of FOXA2 in nude *mice*. Collectively, FTO may be an epigenetic modification target for the treatment of gastric cancer.

#### 3.1.9. FTO in Bladder Cancer

FTO played an oncogenic role in bladder cancer in an m6A-dependent manner [56,57]. Bioinformatic analyses and Western blotting assays showed that FTO was highly expressed in bladder cancer tissues and bladder cancer cells. Moreover, FTO stimulated tumor growth of bladder cancer in vivo and in vitro. Tao et al. illustrated that FTO could increase the expression level of metastasis-associated lung adenocarcinoma transcript 1 (MALAT1) mRNA through m6A demethylation [56]. Further mechanism study demonstrated that FTO promoted the tumorigenesis of bladder cancer by suppressing microRNA miR-384 and inducing mal T cell differentiation protein 2 (MAL2) expression. In the same year, a study by Zhou found that FTO facilitated bladder cancer progression via the FTO/miR-576/cyclin-dependent kinase 6 (CDK6) axis [57]. RNA immunoprecipitation (RIP) assay revealed that FTO mediated miRNA synthesis of miR-576 by regulating the maturation of primiR-576. Additionally, CDK6 was identified as a direct target of miR-576, which could be down-regulated by it. In bladder cancer tissues, the protein level of FTO was positively correlated with CDK6 and negatively related to miR-576. These findings indicated the possibility of FTO as a diagnostic or prognostic biomarker in bladder cancer.

#### 3.1.10. FTO in Esophageal Squamous Cell Carcinoma (ESCC)

Cui et al. discovered that FTO cooperated with LncRNA LINC00022 to promote the tumorigenesis of ESCC by upregulating LINC00022 expression [58]. Clinically, they found that LINC00022 was significantly elevated in primary ESCC samples and correlated with poor prognosis in ESCC patients. Mechanistically, FTO reduced the m6A modification of LINC00022 and promoted its expression, thereby accelerating the proliferation of ESCC cells. However, the genetic addition of YTHDF2 led to a decrease in LINC00022 levels in ESCC cells. On the whole, research demonstrated that FTO-mediated up-regulation of LINC00022 drives ESCC progression in a YTHDF2-dependent manner.

#### 3.1.11. FTO in Multiple Myeloma (MM)

A study has proven that FTO was associated with the progression and metastasis of MM [59]. Transcriptome array analysis indicated that FTO was at a higher level in patients with multiple myeloma compared to healthy subjects. Moreover, FTO functionally increased the expression of heat shock factor 1 (HSF1), a reported metastasis-promoting gene in melanoma, which facilitated the proliferation, migration, and invasion of MM cells [60]. The mechanism by which FTO regulated HSF1 was to block YTHDF2-mediated RNA degradation by eliminating m6A modification. However, inhibition of FTO with MA2 decreased the expression of HSF1 and its target genes in *mice*. Additionally, combination treatment with MA2 and bortezomib, a first-line chemotherapeutic agent for MM, exhibited stronger synergistic cytotoxic effects on MM occurrence and extramedullary metastasis in vivo. These results suggest the FTO-HSF1 axis could be a potential therapeutic target in MM.

### 3.2. ALKBH5 Participates in the Biological Processes of Various Diseases

#### 3.2.1. ALKBH5 in Breast Cancer

ALKBH5 is the second discovered m6A demethylase, of which m6A is the only known catalytic substrate [10]. It is well documented that ALKBH5 can be induced by hypoxia-inducible factor 1α (HIF-1α) under hypoxia conditions [14]. Zhang et al. recently showed hypoxia-induced breast cancer stem cells (BCSCs) enrichment by inducing expression of ALKBH5 [61]. Notably, the BCSC phenotype is characterized by several core pluripotency factors, such as Nanog homeobox (NANOG), which is crucial for maintaining cancer stem cells [62]. In vitro experiments showed that high expression of ALKBH5 induced by hypoxia improved NANOG mRNA’s stability and expression by eliminating the m6A modification of NANOG 3′-UTR. In addition, in vivo experiments have shown that knockdown of ALKBH5 expression impaired tumor formation and reduced BCSC population in breast tumors.

#### 3.2.2. ALKBH5 in Glioblastoma

ALKBH5 is required for maintaining tumorigenicity of GSCs by sustaining transcription factor Forkhead Box M1 (FOXM1) expression [63]. FOXM1, a transcription factor gene, is highly expressed in cancer and is indispensable for the self-renewal and tumorigenesis of GSCs [64,65]. Mechanism analysis revealed that ALKBH5 eliminated m6A modification on 3′-UTR of FOXM1 pre-mRNA accompanied by increasing protein levels of FOXM1. Further, in vivo examinations showed that depletion of ALKBH5 significantly inhibited brain tumor formation. In contrast, this growth inhibition was reversed after ectopic expression of the FOXM1 coding sequence, leading to brain tumor growth. Another team recently demonstrated an important role for ALKBH5 in promoting radioresistance and invasiveness of GSCs [66]. On the one hand, ALKBH5 drove radioresistance by increasing the expression of genes involved in homologous recombination. On the other hand, ALKBH5 upregulated Yes-associated protein 1 (YAP1) expression, thereby contributing to the aggressiveness of GSCs.

#### 3.2.3. ALKBH5 in AML

Recent advances revealed that ALKBH5 played a critical role in promoting tumorigenesis in AML [67]. Primarily, analysis of the gene expression profiling datasets showed that ALKBH5 is highly expressed in various subtypes of AML. Interestingly, TCGA AML database analysis also proved that higher expression of ALKBH5 was related to shorter overall survival in AML patients. Moreover, this finding confirmed that ALKBH5 is indispensable for the self-renewal ability of leukemia stem/initiating cells (LSCs/LICs). The underlying mechanism through which ALKBH5 functioned in AML was carried out by regulating the expression of transforming acidic coiled-coil-containing protein 3 (TACC3). It has been documented that TACC3 played a critical oncogenic role in various cancers and was required for cancer stem cells self-renewal [68]. In this case, ALKBH5 improved the stability of TACC3 mRNA, thus promoting its expression in an m6A-erase manner. These data highlight the role that ALKBH5 played in promoting the tumorigenesis role through the ALKBH5-m6A-TACC3 axis in AML.

#### 3.2.4. ALKBH5 in Ischemic Heart Disease

Abnormal autophagy is associated with many diseases, especially cardiovascular disease [69]. Song et al. recently provided evidence that reversible m6A methylation played a role in the autophagy of cardiomyocytes [70]. Their primarily finding was that hypoxia/reoxygenation (H/R)-treated cardiomyocytes induced m6A levels in total RNA and that METTL3 was the main factor that caused elevated m6A modification. Then, increased METTL3 in H/R-treated cardiomyocytes inhibited autophagic flux and promoted apoptosis. Furthermore, they demonstrated that transcription factor EB (TFEB) was a direct target of METTL3. At the mechanism level, METTL3 methylated TFEB mRNA at the two m6A sites of 3′-UTR. It subsequently facilitated the binding of RNA-binding protein heterogeneous ribonucleoprotein D (HNRNPD) to TFEB pre-mRNA, resulting in its degradation. In contrast, the m6A demethylase ALKBH5 showed an opposite effect on TFEB. Interestingly, further research confirmed that TFEB regulated METTL3 and ALKBH5 in an opposite orientation, i.e., increased ALKBH5 and decreased METTL3. This association between METTL3/ALKBH5 and autophagy implied their potential as therapeutic targets for treating ischemic diseases.

#### 3.2.5. ALKBH5 in Lung Cancer

Additionally, recent papers have shown that ALKBH5 plays diverse roles in lung cancer. Chao et al. found that intermittent the hypoxia-induced high expression of ALKBH5 contributed to the proliferation and invasion of lung adenocarcinoma cells [71]. The mechanism by which ALKBH5 is involved relies on its m6A demethylation activity to induce FOXM1 expression. As previously reported, FOXM1 expression was essential for cancer progression [72]. Further experiments indicated that ALKBH5 exerted its pro-tumor role in lung tissue through upregulating FOXM1. Moreover, Zhu et al. demonstrated that ALKBH5 could accelerate the malignant progression of non-small cell lung cancer (NSCLC) [73]. RNA immunoprecipitation sequencing (RIP-Seq) identified TIMP metallopeptidase inhibitor 3 (TIMP3) as a direct target of ALKBH5. This work confirmed that ALKBH5 was involved in the NSCLC oncogenesis progress by reducing the mRNA stability and protein synthesis of TIMP3 depending on m6A demethylation activity. On the contrary, recent literature by Jin et al. discovered that ALKBH5 inhibited tumor growth and metastasis by decreasing Yes-associated protein (YAP) expression in NSCLC [74]. On the one hand, ALKBH5 inhibited YAP expression by removing m6A methylation of YAP pre-mRNA. On the other hand, ALKBH5 reduced YAP activity by modulating miR-107/large tumor suppressor kinase 2 (LATS2) depending on *human* antigen R (HuR). Collectively, the specific mechanisms of ALKBH5 in lung cancer still need to be explored in future studies.

#### 3.2.6. ALKBH5 in Epithelial Ovarian Cancer

A study by the Zhu group proposed that ALKBH5 promoted epithelial ovarian cancer by controlling autophagy flux [75]. On the one hand, due to activating the EGFR-PIK3CA-AKT-mTOR signaling pathway, ALKBH5 suppressed autophagy to improve the proliferation and invasion of *human* ovarian cancer cells. On the other hand, because of the m6A demethylation capability, ALKBH5 stabilized Bcl-2 mRNA and promoted the formation of the Beclin1–Bcl-2 complex, which resulted in the inhibition of autophagy. Moreover, the high expression of ALKBH5 was found in cisplatin-resistant epithelial ovarian cancer cells to promote cell proliferation and chemoresistance to cisplatin in vivo and in vitro [76]. Research demonstrated that ALKBH5 formed a positive regulation loop with homeobox A10 (HOXA10), thereby maintaining the overexpression of ALKBH5 and HOXA10. Collective results show that ALKBH5 upregulation significantly decreased m6A abundance of Janus kinase 2 (JAK2) mRNA, thus increasing its mRNA expression through inhibiting YTHDF2-mediated mRNA degradation. Notably, they discovered that ALKBH5 overexpression promoted tumor growth and chemoresistance to cisplatin in epithelial ovarian cancer by activating the JAK2/STAT3 pathway, consistent with previous studies which reported that the activation of the JAK2/STAT3 signaling pathway contributes to pro-tumor effect and chemotherapy resistance in several cancer types [77,78,79]. These results indicate that ALKBH5 could be a potential therapeutic target in epithelial ovarian cancer.

#### 3.2.7. ALKBH5 Serves as Pancreatic Cancer (PC) Suppressor

Recent advances have defined ALKBH5 as a suppressor in PC [80,81]. Guo et al. initially investigated the expression of ALKBH5 in PC tissues and found that the expression of ALKBH5 significantly decreased more than that of the corresponding noncancerous tissues [80]. Conversely, the over-expression of ALKBH5 dramatically suppressed proliferation, migration, and invasion of PC cells, and reduced tumor volume in the PC xenograft model. Then, they found that ALKBH5 acted on transcriptome regulation and identified period circadian regulator 1 (PER1) as a downstream target of ALKBH5. The mechanism involved decreased ALKBH5 which upregulated the m6A enrichment of PER1 mRNA and resulted in YTHDF2-mediated mRNA degradation, consistent with the previously reported reduction in PER1 expression in PC [82]. In short, this paper demonstrated that ALKBH5 functioned as a tumor suppressor in PC through modulating transcriptional fate via the ALKBH5/m6A/YTHDF2/PER1 axis. In addition, Tang et al. also showed that the expression of ALKBH5 was reduced in the gemcitabine-treated patient-derived xenograft (PDX) model [81]. In comparison, over-expression of ALKBH5 suppressed the proliferation, migration, and invasion of PC cells and reversed the chemotherapy resistance of pancreatic ductal adenocarcinoma (PDAC) cells. Further experiments showed that ALKBH5 over-expression inactivated Wnt signaling by increasing Wnt inhibitory factor 1 (WIF-1) expression after erasing m6A sites in the 3′-UTR of WIF-1 mRNA. These findings highlight the possibility of an m6A eraser-based approach for the diagnosis and treatment of PC.

#### 3.2.8. ALKBH5 Suppresses Malignancy of Hepatocellular Carcinoma (HCC)

Additionally, Chen et al. proposed that ALKBH5 functioned as a tumor suppressor of HCC via an m6A-dependent and IGF2BP1-associated pattern [83]. Research showed that ALKBH5 was down-regulated in HCC patients, and the low expression of ALKBH5 predicted a poor prognosis of HCC. Moreover, the inhibition of ALKBH5 promoted HCC proliferation and accelerated invasion/metastasis in vitro and in vivo. They verified the anti-oncogenic role of ALKBH5 in HCC. They also identified LY6/PLAUR domain-containing 1 (LYPD1) as a direct target of ALKBH5, and ALKBH5-mediated m6A demethylation blocked IGF2BP1 binding to m6A-containing mRNAs, thereby reducing the stability of LYPD1. It is noteworthy that LYPD1 was subsequently demonstrated to have the ability to induce oncogenic behaviors in HCC. Collectively, this study addresses the important role of the ALKBH5/LYPD1 axis in HCC progression and provides novel insights into therapeutic strategies for HCC.

#### 3.2.9. ALKBH5 in Osteogenesis and Osteosarcoma

Recent papers proposed that ALKBH5 was beneficial in osteogenesis [84,85]. Wang et al. discovered that both ALKBH5 and m6A-containing bone morphogenetic protein 2 (BMP2) were upregulated in the ligamentum flavum cells. During the development of ossification of the ligamentum flavum (OLF), over-expressed ALKBH5 induced BMP2 expression and activated the AKT signaling pathway, thereby promoting the osteogenesis of ligamentum flavum cells [84]. In contrast, METTL3 was a suppressor in the progression of osteogenesis by activating myeloid differentiation factor 88 (MYD88)-mediated NF-κB activity [85]. The upregulation of MYD88 which activated the NF-κB pathway to suppress osteogenesis was confirmed in previous studies. In short, METTL3 promoted the expression of MYD88 through increasing m6A methylation, whereas ALKBH5 had the opposite effect.

Surprisingly, Chen et al. recently found that ALKBH5 was beneficial in osteosarcoma tumorigenesis [86]. The RNA immunoprecipitation assay and RNA pull-down assay confirmed that ALKBH5 interacted with plasmacytoma variant translocation 1(PVT1), an oncogenic long noncoding RNA (lncRNA), thereby inducing its expression. In particular, ALKBH5 removed the m6A modification of the PVT1 transcript and prevented YTHDF2-mediated degradation, thus increasing PVT1 expression. This paper demonstrated that the tumor-promoting effect of ALKBH5 in osteosarcoma was partly through the regulation of PVT1, which was consistent with previous reports of an oncogenic role of PVT1 in cancers. [87].

#### 3.2.10. ALKBH5 in Other Diseases

Li et al. recently examined the expression of ALKBH5 in placental villous tissue from recurrent miscarriage (RM) patients [88]. They found that highly expressed ALKBH5 impaired trophoblastic cell invasion in *human* trophoblasts. Interestingly, ALKBH5 downregulated the stability of cysteine-rich 61 (CYR61) mRNA in an m6A-dependent manner. According to previous researchers, CYR61 played a critical role in the progression of embryogenesis. They inferred from this work that ALKBH5 regulated the pathogenesis of RM by regulating the expression of CYR61.

Additionally, Zhang et al. recently claimed that ALKBH5 was involved in gastric cancer [89]. Both ALKBH5 and the lncRNA nuclear paraspeckle assembly transcript 1 (NEAT1) were highly expressed in gastric cancer cells and gastric cancer tissues. ALKBH5 was beneficial to the expression of NEAT1 by eliminating m6A modification. Moreover, over-expressed NEAT1 combined with the enhancer of zeste homologue 2 (EZH2) to upregulate the expression of downstream genes of EZH2, thereby promoting gastric cancer invasion and metastasis. These results illustrate that ALKBH5 could be a feasible therapeutic target for these diseases.

### 3.3. N6-methyladenosine (m6A) Demethylases Play an Essential Role in Drug Therapy Responses

#### 3.3.1. m6A Demethylases in Chemotherapy Resistance

Recently, Zhou et al. confirmed that FTO played a role in facilitating the chemo-radiotherapy resistance of cervical squamous cell carcinoma (CSCC) [90]. They initially demonstrated that FTO was highly expressed in CSCC tissue and was beneficial to chemo-radiotherapy resistance of CSCC. In addition, FTO stimulated β-catenin expression by erasing m6A modification of β-catenin mRNA. Further research indicates that FTO may contribute to chemo-radiotherapy resistance of CSCC might through inducing the β-catenin expression and subsequently activating excision repair cross-complementation group 1 (ERCC1). Moreover, treatment of CSCC cells with the FTO inhibitor MA2 (Figure 4) improved the chemo-radiotherapy sensitivity.

Shriwas et al. recently reported that *human* RNA helicase DEAD-box helicase 3 (DDX3) was involved in cisplatin resistance in oral squamous cell carcinoma (OSCC) by modulating FOXM1 and NANOG expression via increasing ALKBH5 expression [91]. Importantly, DDX3 expression was elevated in cisplatin-resistant cells and chemo-therapy non-responder tumors. It has been highlighted that an enhanced population of cancer stem cells (CSCs) contributes to chemoresistance and recurrence, and that ALKBH5 promoted CSC property through increasing FOXM1 and NANOG [61,63]. In this case, they confirmed that the specific inhibition of DDX3 decreased the CSC population in chemoresistant cells and significantly suppressed FOXM1 and NANOG expression in an ALKBH5-m6A-dependent manner. In summary, these findings provide new insights for studying the role of m6A demethylase in chemotherapy resistance.

#### 3.3.2. ALKBH5 in Cancer Immunity

Recent research has illustrated the functions and mechanisms of ALKBH5 in cancer immunotherapy. During cancer development, tumor cells evade immune surveillance by expressing inhibitory checkpoint molecules, which is a major mechanism for suppressing immune responses [92]. For example, programmed cell death 1 ligand 1 (PD-L1), a main inhibitory immune checkpoint molecule on tumor cells, contributes to immune evasion by binding to programmed death receptor-1 (PD-1) on T cells [92]. A study found that ALKBH5 was involved in suppressing antitumor T-cell immunity in intrahepatic cholangiocarcinoma by upregulating PD-L1 expression [93]. This work also demonstrated that ALKBH5 reduced the m6A abundance of PD-L1 mRNA, thereby inhibiting the YTHDF2-mediated mRNA degradation.

Li et al. proposed that ALKBH5 played an important role in the resistance to immune checkpoint blockade therapy [94]. Importantly, ALKBH5 deletion enhanced the efficacy of anti-PD-1 therapy and significantly prolonged the survival of ALKBH5-deficient tumor-bearing *mice*. Moreover, ALKBH5 deletion increased m6A abundance in mRNAs, which promoted protein synthesis of several target genes, including monocarboxylate transporter 4 (Mct4)/solute carrier family 16 member 3 (Slc16a3). Further investigation showed that ALKBH5 modulated Mct4 expression and induced lactate content, thereby reducing immune cell populations in the tumor microenvironment during GVAX vaccination and anti-PD-1 antibody therapy. In addition, another team validated that tumor-intrinsic ALKBH5 was responsible for the recruitment of tumor-associated macrophage (TAM) and immunosuppressive phenotypes under hypoxic conditions in glioblastoma multiforme [95]. Interestingly, hypoxia-induced ALKBH5 stabilized lncRNA NEAT1 through m6A demethylation, which subsequently induced the expression and secretion of C-X-C motif chemokine ligand 8 (CXCL8)/interleukin-8 (IL8). CXCL8, a cytokine encoding gene in *humans*, has been widely studied in cancer cells and TAM recruitment [96]. Mechanically, NEAT1 modulated paraspeckle assembly, which in turn induced the relocation of the splicing factor proline- and glutamine-rich (SFPQ) protein from the CXCL8 promoter, ultimately upregulating CXCL8 expression [95].

## 4. Inhibitors

### 4.1. Strategies Used for Developing m6A Demethylases Inhibitors

Since FTO and ALKBH5 rely on cofactors 2OG and Fe^2+^ for their m6A demethylation activity, early studies focused on screening a series of 2OG analogues and related compounds as their inhibitors [97]. Structure-based virtual screening of different compound libraries was an important way to obtain potent FTO/ALKBH5 inhibitors [30,98,99,100,101,102]. Interestingly, a high-throughput fluorescence polarization (FP) assay was performed for compounds that competed with FTO/ALKBH5 for binding to m6A-containing single-stranded nucleic acids, and meclofenamic acid (MA) was found to be a selective inhibitor of FTO over ALKBH5 [103]. Later on, Svensen and Jaffrey reported an approach to identify FTO inhibitors by using a fluorometric RNA substrate based on broccoli aptamer [104]. Das and co-workers designed a multi-protein dynamic combinatorial chemistry (DCC) system for screening FTO inhibitors [105]. More recently, Zhang et al. developed a single quantum dot-based Förster resonance energy transfer (FRET) nanosensor for FTO inhibitor screening [106]. Chang’s team identified several types of compounds that inhibit FTO activity through fluorescence quenching and molecular modeling studies [107,108,109]. Moreover, combining the information from crystal structures of ligand–protein complexes and structure-based drug designs was also an efficient approach to discover potent inhibitors with distinct chemical scaffolds [110,111,112].

### 4.2. FTO Inhibitors

#### 4.2.1. Metal-Chelating Inhibitors

Since the 2OG dependent oxygenases shared the same protein-folding pattern, conserved cofactor, and substrate-binding sites, several 2OG oxygenases generic inhibitors have been shown to suppress FTO demethylation in vitro. These chemical inhibitors included 2OG, as well as pyridyl-, hydroxyquinoline-, and isoquinoline-based compounds [39,97]. Although they showed the different compound scaffolds, these inhibitors were all located in the 2OG binding pocket and chelated the metal ion in a bidentate manner. For instance, the 2OG competitive inhibitor 2-hydroxyglutarate (R-2HG, compound **1**, Figure 3) has been identified as an FTO inhibitor, which displayed anticancer activity in leukemia and glioma [39,113]. N-oxalylglycine (NOG, compound **2**, Figure 3), pyridine-2,4-dicarboxylate (2,4-PDCA, compound **3**, Figure 3), and compound **4** (Figure 3) inhibited FTO demethylation in vitro, the IC_50_ values of which were 44, 8.3 and 15 μM, respectively [97].

To better understand ligand–protein interactions, we generated 2D protein-ligand interaction diagrams from crystal structure complexes retrieved from the Protein Data Bank using MOE software. The 2D protein-ligand interaction diagrams were processed in four steps: (1) load the PDB file of the crystal complex into the MOE software; (2) rotate the crystal structure to a suitable angle and click “Compute” and “Ligand Interactions” buttons to create 2D diagrams; (3) change the “Legend” dropdown to “Rendering Options”, increase the residue size to 1.8 angstroms, and click “Apply”; and (4) save the diagram as an image in the TIF format with default parameters. Structures for FTO in complexes with NOG (PDB ID: 4IDZ) and 2,4-PDCA (PDB ID: 4IE0) showed that both of them are bound to metal ions (Figure 5A,B). Moreover, they further interacted with residues Arg316, Ser318, and Tyr295 of the side chains. In the complex of FTO with 4 (PDB ID: 4IE5), the pyridine ring of 4 nearly reached the substrate-binding site of FTO, which might spatially compete with the catalytic substrate (Figure 5C). Moreover, 8-QH (compound **5**, Figure 3) was a relatively potent FTO inhibitor with an IC_50_ value of 3.3 μM. The crystal structure of FTO-8-QH (PDB ID: 4IE4) showed that 8-QH doubly chelated the Zn^2+^ ion with hydroxyl and nitrogen of the hydroxyquinoline in a similar way to NOG (Figure 5D). IOX3 (compound **6**, Figure 3) and FG-4592 (compound **7**, Figure 3) were known as prolyl-hydroxylase inhibitors [114], which also showed good inhibitory activity against FTO with IC_50_ of 2.8 and 9.8 μM, respectively [97,115]. The crystal structure of FTO-IOX3 (PDB ID: 4IE6) indicated that its chlorine atom of the isoquinoline group reached the substrate-binding site (Figure 5E).

In 2012, Chen et al. identified the natural product rhein (compound **8**, Figure 3) (IC_50_ = 21 μM) as a competitive substrate inhibitor of FTO [98]. Further, rhein was the first discovered cell-active FTO inhibitor, which could inhibit cellular FTO demethylase activity. In molecular modeling of FTO-rhein (PDB ID: 4IE7), rhein occupied the binding sites of 3-meT, 2OG, and Fe^2+^. It is important to mention that this special structure blocked the binding of m6A containing ssDNA/ssRNA substrates to FTO (Figure 5F). Compound **9a** (Figure 3) acted as a selective inhibitor of FTO (IC_50_ = 0.6 μM) compared to ALKBH5 (IC_50_ = 96.5 μM) and other AlkB subfamilies [116]. To view the superimposition from an FTO-3-meT-NOG (PDB ID: 3LFM) structure with that of FTO-**9a** (PDB ID: 4CXW), **9a** occupied both 2OG and nucleotide acid binding sites (Figure 5G). The fumarate hydrazide of **9a** was bound in the same combination as NOG, while the 4-benzyl pyridine side-chain sat in the nucleotide-binding site. Inferentially, the interaction between the pyridine nitrogen atom of **9a** and Glu234 of FTO was the key factor for the high binding selectivity of FTO. In contrast, in other AlkB subfamilies, it was significantly weakened and even disappeared. In particular, both compound **9a** and its ethyl ester derivative **9b** (Figure 3) showed low cytotoxicity and significantly increased the global level of m6A in HeLa cells. Shishodia et al. used knowledge of the interaction of FTO with 2OG and substrates to design synthetic FTO inhibitors, of which compound **10** (IC_50_ = 1.5 μM, Figure 3) exhibited the best inhibitory activity [110].

Compound MO-I-500 (compound **11**, Figure 3), a dihydroxyfuran sulfonamide [117], was the first identified as an FTO inhibitor, which displayed anticonvulsant activity. In the superposition of the MO-I-500 to NOG-FTO complex (PDB ID: 3LFM), this compound is located at the 2OG active site, and the hydroxyl oxygens of dihydroxyfuran chelated with the metal ion in opposite directions. The molecule MO-I-500 displayed anticonvulsant activity in the 6 Hz *mouse* model at a nontoxic dose, increased the total m6A level of cellular RNA, and altered the production of relative microRNAs. Through using a multi-protein DCC strategy, compound **12** (Figure 3) was identified as a FTO (IC_50_ = 2.6 μM) selective inhibitor, in comparison with ALKBH5 (IC_50_ = 201.3 μM) [105]. The structural model of FTO-**12** revealed that compound **12** coordinated with Fe^2+^ in a bidentate manner, which was further stabilized by a combination of hydrogen-bonding and salt bridge interactions with Arg96, Arg319, Tyr295, and Ser318 of side chains from FTO. Two compounds **13a** (IC_50_ = 1.46 µM, Figure 3) and **13b** (IC_50_ = 28.9 µM, Figure 3) were defined as FTO inhibitors through a virtual screening on the ZINC compound library [99]. Molecular docking calculations revealed specific interactions between the amino acid residues of the FTO proteins Asp233, Tyr106, Glu234, Arg96, and Arg322, as well as two compounds. Importantly, compounds **13a** and **13b** are the first FTO inhibitors demonstrated to support the survival and rescue dopamine neurons from growth factor deprivation-induced apoptosis in vitro.

#### 4.2.2. Substrate Competitive Inhibitors

Meclofenamic acid (MA) (compound **14a**, Figure 4) and its derivatives were determined to be substrate competitive selective inhibitors of FTO [103,118,119]. When structural superimposition of the complexes of FTO-MA (PDB ID: 4QKN) and FTO-3-meT (PDB ID: 3LFM) was accomplished, in this case, MA partially covered the binding site of 3-meT in an L shape. In addition, there were stable hydrophobic interactions between a part of the FTO NRL and the carboxyl acid substituent of MA (Figure 5H) [103]. However, these hydrophobic interactions did not appear in the NRL of ALKBH5, which reduced the binding of MA to ALKBH5. MA2 (compound **14b**, Figure 4), an ethyl ester derivative of MA, was a cell-active inhibitor of FTO, which could enhance the overall level of m6A in HeLa cells. Inspired by the specific binding of MA to FTO, fluorescein (compound **15a**, Figure 4) and its derivatives, i.e., FL2 and FL2-DZ (compound **15b** and compound **15c**, Figure 4), were explored as FTO inhibitors with IC_50_ = 3.23 μM, 1.72 μM, and 4.49 μM, respectively. In FTO fluorescein’s (PDB ID: 4ZS2) crystal, fluorescein sat in the nucleotide-binding site of FTO, which was similar to MA (Figure 5I). Among them, FL2-DZ could selectively inhibit the demethylation of FTO. FL2-DZ also showed specific photo-affinity labeling of intracellular FTO because of the diazirine unit [119]. Thus, these fluorescein derivatives have dual functions of inhibiting FTO activity and labeling FTO. More recently, selective inhibitors FB23 (compound **16a**, Figure 4) and FB23-2 (compound **16b**, Figure 4) were synthesized by extending the dichloride-substituted benzene of MA. They were more efficient with IC_50_ values of 0.06 μM and 2.6 μM, respectively [118]. In the FTO-FB23 crystalline complex (PDB ID: 6AKW), FB23 occupied the entire binding position of MA in a similar L shape (Figure 5J). For FB23, the phenyl carboxylic acid substituent of MA was retained, forming several hydrophobic interactions with the nucleotide recognition cap. Hence, it showed the specific recognition capability of FTO compared to ALKBH5. Moreover, several hydrogen bonds were found between nitrogen or oxygen in the heterocyclic ring of FB23 and Glu234 of FTO, which was beneficial for the FB23 inhibitory activity of FTO. In vitro and in vivo research confirmed that FB23-2 improved the anti-proliferative activity of AML cell line cells and inhibited primary AML LSCs in *mouse* models.

A series of benzene-1,3-diol derivatives were identified as selective inhibitors of FTO. They were N-CDPCB (compound **17**, Figure 4) [111], CHTB (compound **18**, Figure 4) [120], and radicicol (compound **19**, Figure 4) [100]. IC50 values of N-CDPCB, CHTB and radicicol were 4.95 μM, 39.24 μM and 16.04 μM, respectively. In the crystal of compounds FTO-N-CDPCB (PDB ID: 5DAB) and N-CDPCB was sandwiched between the β-sheet and the L1 loop of FTO at the extension of the 2OG binding site (Figure 5K) [111]. In addition, the chlorine group was crucial for strengthening the N-CDPCB-FTO complex [111]. According to the binding pocket of N-CDPCB to FTO, a novel binding site was observed, which was partly overlapped with the inhibitor MA, not the 3-meT position. Interestingly, CHTB occupied the entire MA binding site in a similar L-shaped fashion in the crystal of FTO-CHTB (PDB ID: 5F8P) [120]. There were visible interactions between the chlorine atom in the chroman ring and several residues (Val83, Ile85, Leu90, and Thr92) of FTO in the hydrophobic pocket (Figure 5L). A hydrogen bond was also formed between residue Glu234 and the benzene hydroxyl group. Moreover, both N-CDPCB and CHTB were able to increase m6A abundance in total mRNA in 3T3-L1 cells. Inspired by the common structural features of N-CDPCB and CHTB, Chang’s group performed a structure-based virtual screening of compounds containing the 4-Cl-1,3-diol group and identified the natural compound radicicol as an effective FTO inhibitor [100]. Radicicol is bound to FTO and located at a similar cavity in the crystal of the FTO-radicicol complex, compared to N-CDPCB’s in an L-shaped conformation. One of the obvious differences between these two crystal complexes was that the conservative 4-Cl-1,3-diol group was bound to FTO in different orientations.

Additionally, by using Schrödinger software for molecular docking to target the MA binding site of FTO, a study designed and synthesized chemically distinct FTO inhibitors, of which FTO-04 (compound **20**, Figure 4) was identified as a competitive inhibitor of FTO (IC_50_ = 3.39 μM) over ALKBH5 (IC_50_ = 39. 4 μM) [112]. Importantly, research demonstrated that FTO could impair the self-renewal properties of GSCs to inhibit neurosphere formation without altering the growth of *human* neural stem cell neurospheres. Prakash and co-workers synthesized compound **21a** (Figure 4) as a potent FTO selectivity inhibitor (IC_50_ = 0.087 μM) by merging the key fragments of compound **9a** and MA [121]. Moreover, the ester prodrug **21b** of compound **21a** could reduce the viability of AML cells by downregulating MYC and upregulating RARA, which was consistent with previous reports on the anticancer effect of pharmacological FTO inhibition [37,39].

In 2020, Chen and co-workers determined CS1 (compound **22a**, Figure 4) and CS2 (compound **22b**, Figure 4) as potent and selective FTO inhibitors through conducting a structure-based virtual screening [101]. Both CS1 and CS2 displayed a much higher anti-leukemic efficacy in comparison to FB23–2 in vitro and in vivo by modulating the expression of FTO target genes, including MYC, RARA, and ASB2. Moreover, this study also confirmed that CS1 and CS2 reprogramed immune response by reducing immune checkpoint gene expressions, especially leukocyte immunoglobulin-like receptor (LILRB4) [101]. In the same year, diacerein (compound **23**, Figure 4) was identified as an FTO inhibitor by using a single quantum dot-based FRET nanosensor with an IC_50_ value of 1.51 μM [106]. Molecular modeling studies have suggested that diacerein possibly competed with m6A-containing ssDNA for FTO binding through forming hydrogen bonding with the amino acid residues of FTO protein. In addition, researchers validated the anti-proliferation effects of Saikosaponin-d (SsD, compound **24**, Figure 4) in AML by targeting the m6A demethylation activity of FTO [122]. In vitro experiments showed that SsD exhibited good inhibitory activity on FTO demethylation with a low IC_50_ value of 0.46 μM. Importantly, they also demonstrated that SsD could overcome the resistance to tyrosine kinase inhibitors by suppressing FTO-mediated m6A RNA methylation pathways.

#### 4.2.3. FTO Inhibitors with Other Scaffolds

Svensen and Jaffrey found a fluorogenic-methylated substrate for FTO based on the Broccoli aptamer [104]. After demethylation by FTO, this substrate was fluorescent. Subsequently, the fluorescent substrate was utilized to high-throughput screen FTO inhibitors with different chemical structures. As a consequence, a series of novel effective inhibitors were found, including amiloride analogue compound **25a** (Figure 6), methionine derivative **25b** (Figure 6), rhein analogues **25c-1** and **25c-2** (Figure 6), MA analogues **25d** (Figure 6), and other scaffolds (**25e-1**, **25e-2**, **25e-3**, **25e-4**, **25e-5**) (Figure 6) with low IC_50_ values ranging from 0.34 μM to 3.00 μM. In comparison with ALKBH5, **25e-1** and **25e-4** showed selectivity for FTO. Further, **25c-2**, **25e-1**, and **25e-3** were cell-active in inhibiting FTO demethylase activity. These compounds provide new information for the design of more potent FTO inhibitors with new structural scaffolds.

Additionally, through structure-based virtual screening of U.S. Food and Drug Administration (FDA)-approved drugs, Peng et al. discovered that entacapone (compound **26a**, Figure 6) was a substrate, as well as the 2OG cofactor competitive inhibitor of FTO [30]. Entacapone was structurally distinct from any reported inhibitors of FTO, the IC_50_ value of which was 3.5 μM. In the crystal of entacapone bound with FTO (PDB ID: 6AK4), hydrogen bonds could be discovered between the heterotopic hydroxyl group on the nitrocatechol ring with residues from the substrate-binding site (Figure 5M). Additionally, the nitrile group of the compound could chelate with Zn^2+^, which was recently reported in histone demethylase protein–ligand complex cases. Interestingly, the flexible tail of diethyl-propanamide was embedded deeply in the cofactor binding site. Furthermore, compounds **26b** (Figure 6) and **26c** (Figure 6) were designed and synthesized by replacing the flexible diethyl tail of entacapone with alicyclic groups, enhancing the inhibitory activity of FTO with IC_50_ values of 1.2 and 0.7 μM, respectively.

Combining the fluorescence quenching technology, several inhibitors were found to decrease the demethylase activity of FTO, including nafamostat mesylate (compound **27**, Figure 6) [107], clausine E (compound **28**, Figure 6) [108], 2-phenyl-1H-benzimidazole **29** (Figure 6) [109], fluoronucleoside analogue **30** (Figure 6) [123], (s)-hydroxycamptothecin (compound **31**, Figure 6) [124], flavonols **32a** and **32b** (Figure 6) [125], clenbuterol **33** (Figure 6) [126], pyrazole derivative **34** (Figure 6) [127], 1,3-diazaheterocyclic compounds **35a** and **35b** (Figure 6) [128], and 3-substituted 2-aminochromones **36a** and **36b** (Figure 6) [129]. Among them, nafamostat mesylate, clausine E, and compound **29** showed good inhibitory activity against FTO with IC_50_ values of 13.77 μM, 27.79 μM, and 24.65 μM, respectively. Moreover, molecular docking model analysis showed that the affinity bindings between FTO and these molecules were mainly forced by the hydrophobic and hydrogen bonds interactions with residues from the active cavity of FTO, which were similar to the binding modes between FTO and other inhibitors.

### 4.3. ALKBH5 Inhibitors

Recently, several research groups found that 2OG inhibitors showed the inhibiting activity of ALKBH5 demethylase. In these cases, the crystalline complexes of ALKBH5 with different compounds were also obtained to display the direct structural evidence [16,17,18,19]. For instance, NOG and succinate (compound **37**, Figure 7) showed IC_50_ values of 25.85 μM and 30.00 μM, respectively. However, different from FTO, 2,4-PDCA (IC_50_ = 347.2 μM) and citrate (compound **38**, IC_50_ = 488 μM) were moderate inhibitors of ALKBH5 (Figure 7). In the ALKBH5 crystal with NOG, succinate, and 2,4-PDCA (PDB IDs: 4NRP [17], 4NPM [19], or 4NRQ [17]), all of the corresponding inhibitors were located in the 2OG active site of ALKBH5 and chelated with Mn^2+^ (Figure 8A–C). Moreover, according to the overlay of ALKBH5-citrate (PDB entry 4O61) and FTO-citrate (PDB entry 4IE7), the citrate molecule competed with 2OG, and the 2OG binding sites were partially covered by its positions (Figure 8D) [18]. One possible reason was that the residues Ile281 and Tyr195 of ALKBH5 blocked citrate from reaching the 2OG active site. Notably, even if citrate displayed the modest inhibitory activity on the ALKBH family dioxygenases, the discrepancies of citrate binding to FTO and ALKBH5 could provide a strategy to design new types of ALKBH5-specific inhibitors.

The imidazobenzoxazin-5-thione MV1035 (compound **39**, Figure 7) was demonstrated to inhibit ALKBH5 demethylase activity in vitro. At the same time, it significantly reduced the migration and invasiveness of the U87 glioblastoma cell lines [130]. In this case, they reported a potential binding site for MV1035 within ALKBH5. MV1035 overlapped with the catalytic site of the enzyme, specifically near the carboxylate group of 2OG. Recently, a structure-based virtual screening identified two compounds, 2-[(1-hydroxy-2-oxo-2-phenylethyl)sulfanyl]acetic acid(compound **40**, Figure 7)) and 4-{[(furan-2-yl)methyl]amino}-1,2-diazinane-3,6-dione (compound **41**, Figure 7)), which acted as ALKBH5 inhibitors with IC_50_ values of 0.84 μM and 1.79 μM, respectively [102]. This study also demonstrated that these two inhibitors suppressed the cell proliferation of several AML cell lines at low micromolar concentrations, with IC_50_ ranging from 1.38 to 16.5 μM. In addition, some FTO inhibitors also showed similar activity with ALKBH5. For example, amiloride analogue **25a** (Figure 7), rhein analog **25c-2** (Figure 7), and compound **25e-3** (Figure 7) showed inhibitory activity with IC_50_ values of 1.98 μM, 7.13 μM, and 1.24 μM for target ALKBH5, respectively.

## 5. Conclusions

Numerous epigenetic studies in recent decades have revealed the potential of m6A demethylases as therapeutic targets for Human Diseases, including cancers. However, these two demethylases are likely to display different effects on various diseases because they are abundant in different tissues and have a different selection of action sites. Given the above, FTO is intensely related to a range of biomedical processes, including obesity-related diseases, metabolic homeostasis, pro-tumor, self-renewal ability of CSCs, immunotherapy resistance, and chemotherapy resistance. Correspondingly, ALKBH5 shows a clear association with pro-tumor, cancer suppression, self-renewal ability of CSCs, autophagy, chemotherapy resistance, and other diseases. Nevertheless, few studies have reported on the regulation of cellular contexts, such as immunity, DNA damage, autophagy, and apoptosis by m6A demethylases. Thus, future work is certainly required to determine the regulatory genes of each m6A demethylase in various cancers.

At present, reported FTO/ALKBH5 inhibitors mainly focus on 2OG analogs and substrate-competitive inhibitors. Several potent FTO inhibitors have been demonstrated to suppress the proliferation of cancer cells, such as R-2HG, FB23-2, CS1, CS2, and SsD in leukemia cells [39,101,118,122]; MA2 in GSCs [41]; and MO-I-500 in breast cancer cells [45]. Moreover, in vivo studies have also shown that several selective FTO inhibitors can significantly inhibit tumor growth and prolong survival in *mice*. However, few of the currently developed small-molecule FTO inhibitors are available for clinical applications due to mild bioavailability, low sensitivity, and/or poor selectivity. Therefore, much work remains to be carried out in the future to develop more potent FTO inhibitors and to improve their biological functions, inhibitory effects, and the therapeutic potential for *human* disease treatment. In addition, designing and synthesizing FTO inhibitors based on existing small molecules, and discovering FTO inhibitors with distinct frameworks or compounds that bind to novel binding sites, should be considered as an important research strategy in the future work. Additionally, the lack of selectivity was a challenge for ALKBH5 inhibitors. One significant hindrance is its flexible motif 2 modification and unique disulfide bond, resulting in the smaller active pocket and smaller inhibitors. Therefore, identifying new and lead ALKBH5 inhibitors is urgently needed.

In conclusion, this review underlines the recent advances of m6A demethylases in many Human Diseases. However, there are still some issues that should be resolved. Firstly, the underlying mechanisms of m6A demethylases in some cancers are not fully understood. Secondly, some findings have shown that m6A demethylases can be used as therapeutic targets, but specific experiments in clinical trials remain to be conducted. Thirdly, the possibility of m6A demethylase inhibitors for further clinical applications, or in combination with clinical drugs for specific diseases, should be carefully explored.

## Figures and Tables

**Figure 1 ijms-23-05815-f001:**
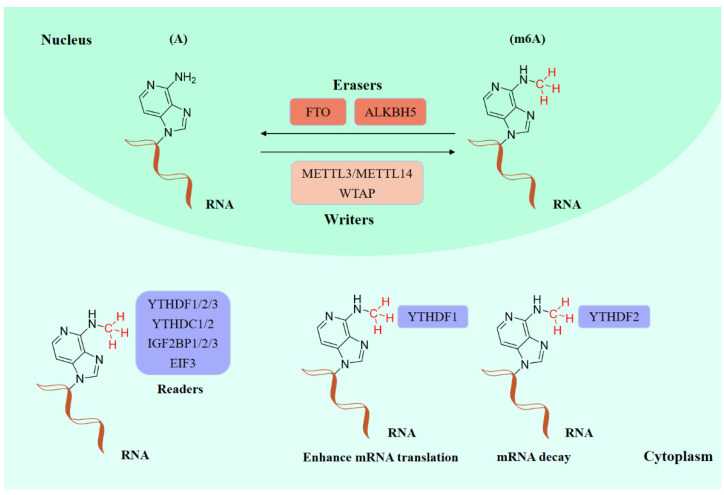
The molecular mechanism of m6A modifications.

**Figure 2 ijms-23-05815-f002:**
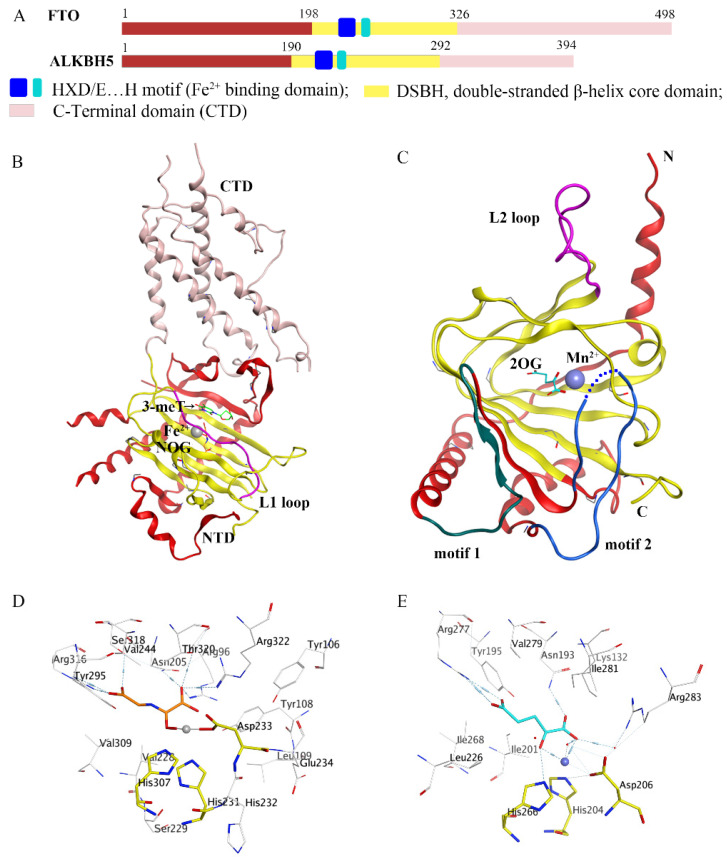
Comparison of fat-mass- and obesity-associated protein (FTO) and alkB homolog 5 (ALKBH5). (**A**) Structural domain representation of FTO and ALKBH5. (**B**) Crystal structure of FTOΔ31-3-meT (PDB ID: 3LFM) generated by MOE software. C-terminal domain (CTD) is colored in pink; N-terminal domain (NTD) is colored in red; the L1 loop is colored in purple; 3-meT is colored in green; N-oxalylglycine (NOG) is colored in orange; Fe^2+^ is colored in grey. (**C**) Crystal structure of ALKBH5_66-292_-2OG (PDB ID: 4NRO) generated by MOE software; the motif 1 region is colored in dark green; the motif 2 region is colored in dark green; the L2 loop is colored in purple; 2-oxoglutarate (2OG) is colored in cyan, Mn^2+^ is colored in light blue; N: N-terminus, C: C-terminus. (**D**,**E**) The detailed interactions of the active center of FTO (PDB ID: 3LFM) and ALKBH5 (PDB ID: 4NRO) generated by MOE software. NOG and 2OG are colored in orange and cyan, respectively; Fe^2+^ and Mn^2+^ are drawn as grey and blue balls, respectively.

**Figure 3 ijms-23-05815-f003:**
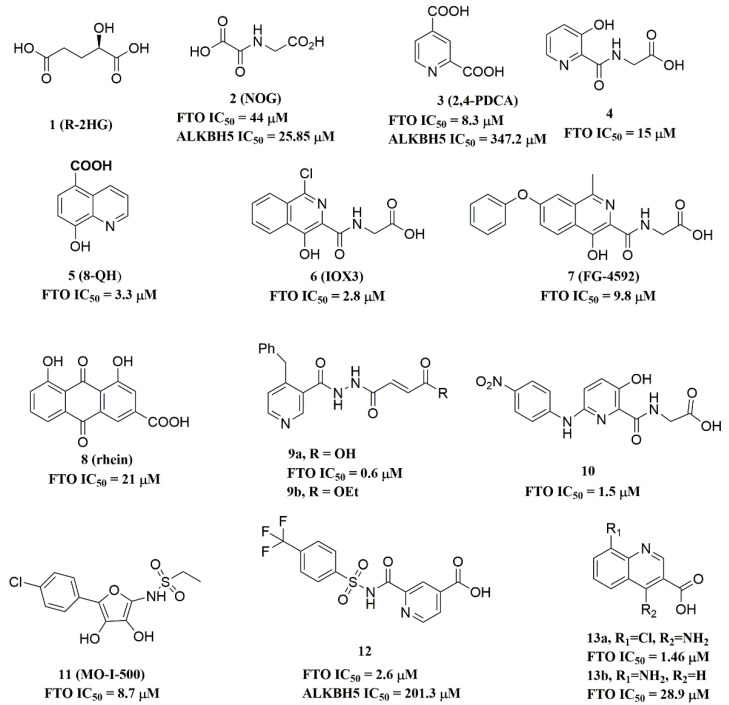
Metal-chelating inhibitors of FTO.

**Figure 4 ijms-23-05815-f004:**
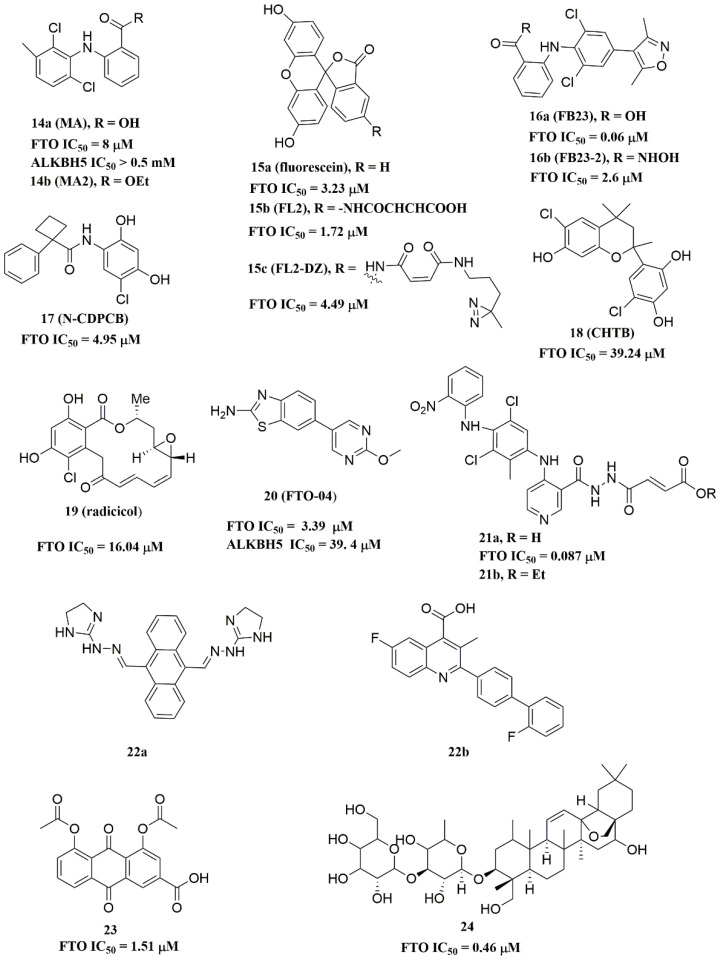
Substrate competitive inhibitors of FTO.

**Figure 5 ijms-23-05815-f005:**
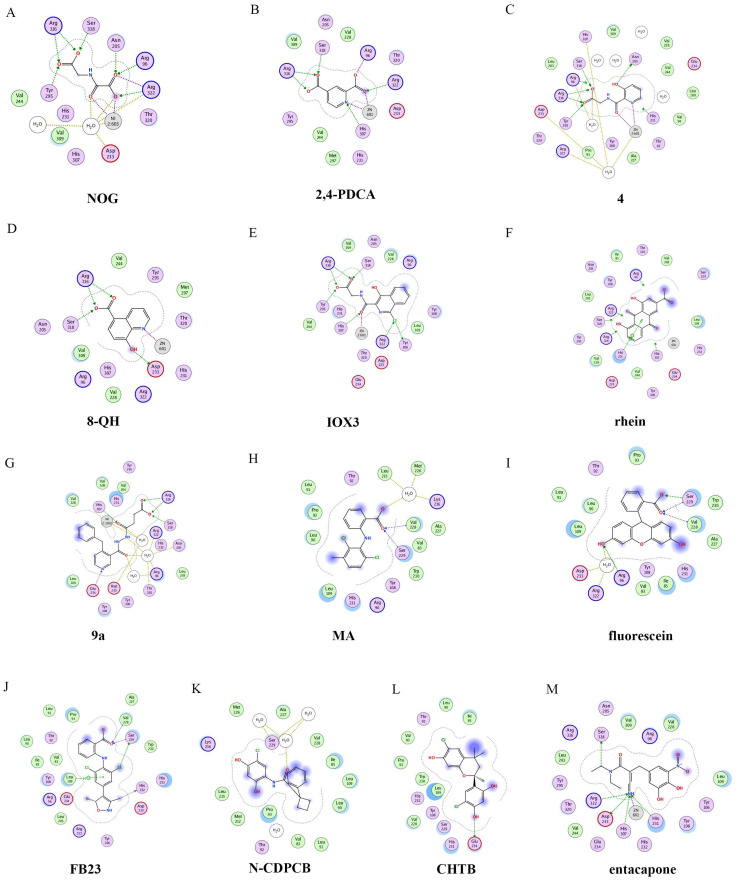
Two-dimensional representation of ligand–protein interactions of FTO inhibitors with FTO protein using MOE software. In MOE, the polar and non-polar residues are shown in pink and green disks; the water molecules are drawn as white circles; the metal ions are shown in grey circles; the hydrogen bonds are indicated by green dotted lines. (**A**–**M**) Ligand interactions of FTO with distinct compounds: (**A**) NOG (PDB ID: 4IDZ); (**B**) pyridine-2,4-dicarboxylate (2,4-PDCA, PDB ID: 4IE0); (**C**) **4** (PDB ID: 4IE5); (**D**) 8-QH (PDB ID: 4IE4); (**E**) IOX3 (PDB ID: 4IE6); (**F**) rhein (PDB ID: 4IE7); (**G**) **9a** (PDB ID: 4CXW); (**H**) Meclofenamic acid (MA**)** (PDB ID: 4QKN); (**I**) fluorescein (PDB ID: 4ZS2); (**J**) FB23 (PDB ID: 6AKW); (**K**) N-CDPCB (PDB ID: 5DAB); (**L**) CHTB (PDB ID: 5F8P); and (**M**) entacapone (PDB ID: 6AK4).

**Figure 6 ijms-23-05815-f006:**
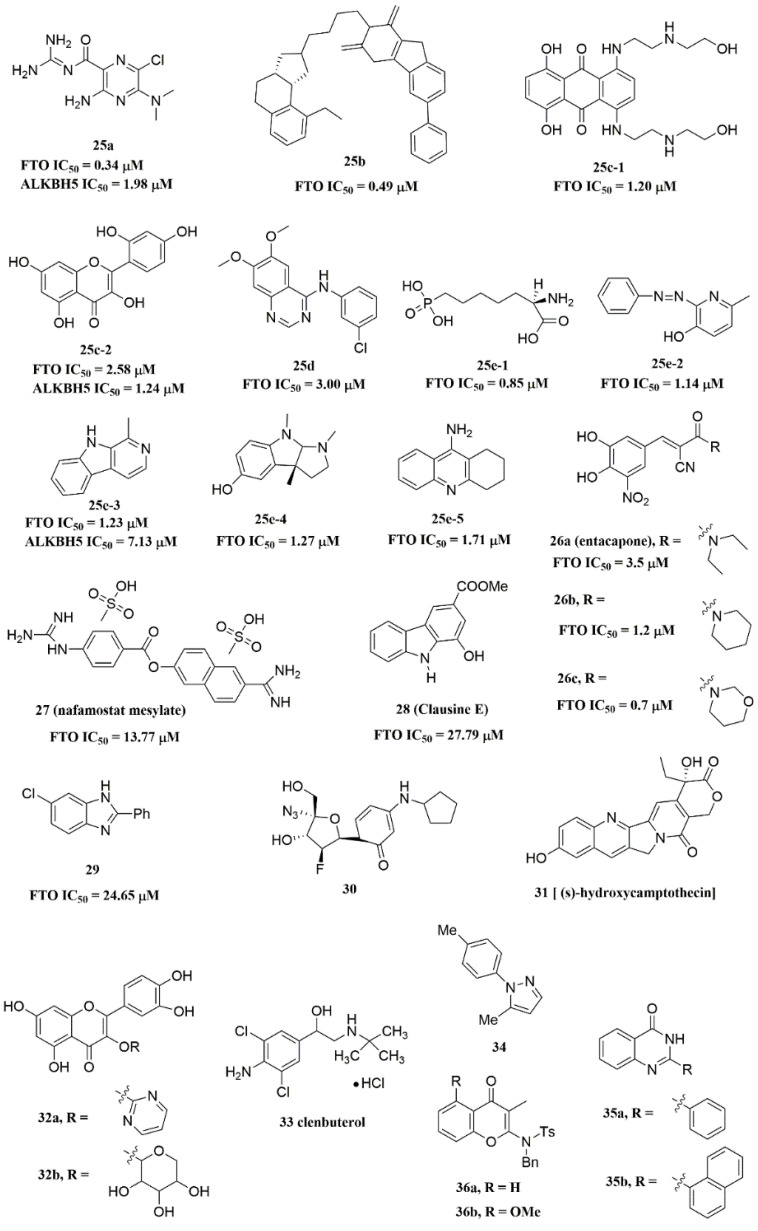
FTO inhibitors with different scaffolds.

**Figure 7 ijms-23-05815-f007:**
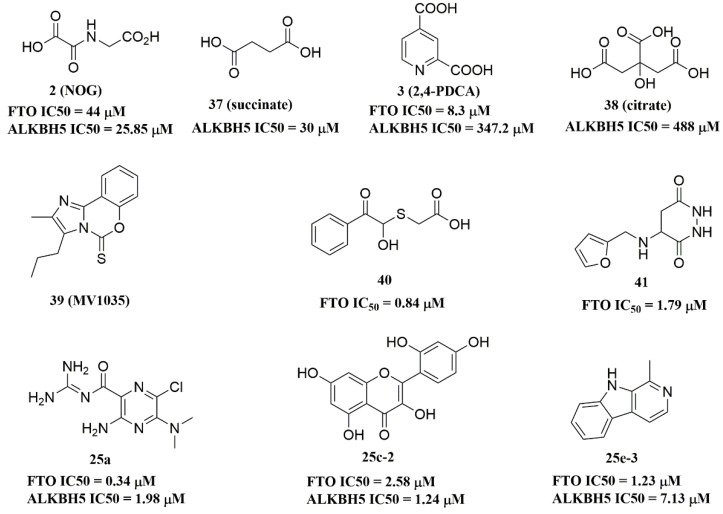
ALKBH5 inhibitors.

**Figure 8 ijms-23-05815-f008:**
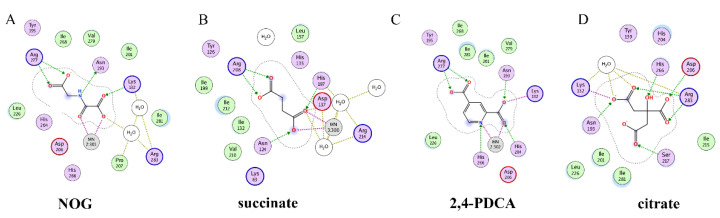
Two-dimensional representation of ligand–protein interactions of ALKBH5 inhibitors with ALKBH5 protein using MOE software. (**A**–**D**) Crystal structures of ALKBH5 with different compounds: (**A**) NOG (PDB ID: 4NRP); (**B**) succinate (PDB ID: 4NPM); (**C**) 2,4-PDCA (PDB ID: 4NRQ); (**D**) citrate (PDB ID: 4O61).

## Data Availability

Not applicable.

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
