# Peer review of "Recent Advances of m6A Demethylases Inhibitors and Their Biological Functions in Human Diseases"

_ijms, 2022, doi:10.3390/ijms23105815_

Round 1

Reviewer 1 Report

Although the manuscript presents an interesting topic, it needs substantial improvement. Therefore, I recommend the authors' team consider the following points during the revision of the manuscript.

  1. The authors should highlight information about the used databases for collecting and or extracting the data (for example, Web of Science, Scopus, Google Scholar,..) and what keywords were used for the literature search along with the period of covering the collected studies. This is to ensure that the paper covers all available recent and relevant studies. All these points could be highlighted, at least, in the introduction section.
  2. Figures 2, 6, and 8. The authors should provide information about the procedure (including molecular docking methods in detail) of preparing ligand-receptor docking complexes. 
  3. Analyzing and discussing the collected data critically. In other words, the reviewed studies and the acquired data should be deeply discussed and interpreted. Unfortunately, the manuscript in its current form looks like a hasty report on m6A demethylases inhibitors without assessing whether the discussed studies are valid. This point should be carefully considered by the authors. 
  4. Since the title of the manuscript focuses on the recent advances in m6A demethylases inhibitors, the authors should add a new section that discusses the current and future strategies/technologies used for developing effective and safe m6A demethylases inhibitors.
  5. Finally, I recommend the authors' team double-check the whole manuscript for grammatical and typing errors. 

Reviewer 2 Report

In the manuscript “Recent advances of m6A demethylases inhibitors and their biological functions in human diseases” the authors give up-to-date information about m6A 16 demethylases - obesity-associated protein (FTO) and alkB homolog 5 (ALKBH5). The structure of these proteins, their involvement in the development of diseases, mainly malignant tumors of various localizations, and their possible inhibitors are discussed. With an overall positive assessment of this review, there are a number of comments to the manuscript.

  1. In section 3.1. "FTO is involved in the regulation of various diseases" has a detailed description of the involvement of FTO in the pathogenesis of various malignant diseases, but there is no section on cardiovascular disease data.
  2. This section has subsection «3.1.5. FTO in nervous system development», which describes the normal development of the nervous system, but it is not a human disease. At the same time, there are no data on neuropsychiatric diseases, including Alzheimer's disease.

Round 2

Reviewer 1 Report

Dear Authors, 

Although the manuscript has been improved, it requires further improvement regarding all figures that present molecular docking complexes (ligand-protein interactions). The authors should provide information about the molecular docking procedures and the validation protocols, otherwise, the presented interactions can not be taken as reliable. The readers should be informed in the manuscript about how did you perform in detail the molecular docking procedures and if they were correctly performed. These points should be considered to validate if the presented figures are correct and reliable.   

Round 3

Reviewer 1 Report

The manuscript has been sufficiently improved.